# Prostaglandin E2 Induces Skin Aging via E-Prostanoid 1 in Normal Human Dermal Fibroblasts

**DOI:** 10.3390/ijms20225555

**Published:** 2019-11-07

**Authors:** Joong Hyun Shim

**Affiliations:** Science & Engineering Bldg, Faculty of Cosmetics and Beauty biotechnology, Semyung University, 65 Semyung-ro, Jecheon, Chungbuk 390-711, Korea; jhshim@semyung.ac.kr; Tel.: +82-43-649-1615; Fax: +82-43-649-1730

**Keywords:** PGE2, EP1, skin-aging, ERK1/2, intracellular calcium

## Abstract

Collagen type I production decreases with aging, leading to wrinkles and impaired skin function. Prostaglandin E2 (PGE2), a lipid-derived signaling molecule produced from arachidonic acid by cyclo-oxygenase, inhibits collagen production, and induces matrix metallopeptidase 1 (MMP1) expression by fibroblasts in vitro. PGE2-induced collagen expression inhibition and MMP1 promotion are aging mechanisms. This study investigated the role of E-prostanoid 1 (EP1) in PGE2 signaling in normal human dermal fibroblasts (NHDFs). When EP1 expression was inhibited by EP1 small interfering RNA (siRNA), there were no significant changes in messenger RNA (mRNA) levels of collagen, type I, alpha 1 (COL1A1)/MMP1 between siRNA-transfected NHDFs and siRNA-transfected NHDFs with PGE2. This result showed that EP1 is a PGE2 receptor. Extracellular signal-regulated kinase 1/2 (ERK1/2) phosphorylation after PGE2 treatment significantly increased by ~2.5 times. In addition, PGE2 treatment increased the intracellular Ca^2+^ concentration in NHDFs. These results indicated that PGE2 is directly associated with EP1 pathway-regulated ERK1/2 and inositol trisphosphate (IP_3_) signaling in NHDFs.

## 1. Introduction

The elderly have typically thin and fragile skin, with increased susceptibility to bruising and impaired wound healing [1]. These aging-related alterations largely reflect collagen fragmentation and reduction. Collagen is the major fibrous component of the dermal extracellular matrix (ECM) [2]. In human skin, collagen fibers account for ~75% of the dry weight of the dermis, while collagen type I makes up 80%–90% of the total collagen. Although mature collagen fibers are stable, with a half-life of ~15 years [3], gradual matrix metallopeptidase (MMP)-mediated ECM degradation occurs during aging [4], resulting in the accumulation of denatured fragments. Therefore, the dermal ECM becomes progressively disorganized, altering the function of skin dermal fibroblasts, during aging [4].

Most fibroblasts represent the major cell type of connective tissue. The main role of fibroblasts is to synthesize the ECM and maintain ECM homeostasis. Fibroblasts attach to the surrounding ECM to form adhesion complexes, which, in turn, act through the cytoskeleton to exert contractile forces. Resistance to these contractile forces produces mechanical forces in the fibroblasts, which primarily determine morphology, cytoskeletal tissue organization, gene expression, and signaling transduction [5,6,7]. Contractile forces, with concomitant mechanical forces, are associated with down-regulation of procollagen production in human skin [2,4,8]. Collagen degeneration causes loss of attachment sites for fibroblasts, decreasing the spreading of fibroblasts, which is observed in the elderly.

Prostaglandin E2 (PGE2), a lipid-derived signaling molecule, is a major prostaglandin found in mammalian skin [9]. Especially, fibroblasts in aging skin display an increase in PGE2 levels [10,11,12]. PGE2 is generated from arachidonic acid by sequential actions of cyclo-oxygenases 1 and 2 (COX1 and COX2) and prostaglandin E synthases 1, 2, and 3 (PGES1, PGES2, and PGES3) [11,12]. Elevated COX2 levels are often consistent with PGES1 induction in various tumor lesions and in response to inflammatory stimuli [13,14,15]. PGE2 is usually synthesized at low levels but significantly increases in skin squamous cell carcinoma, skin inflammatory conditions, such as sunburn, and aging skin [10,16]. PGE2-mediated intracellular signal transduction depends on the binding of cells to one or more prostaglandin E receptors (E-prostanoid 1–4 [EP1–EP4]), which are coupled to different G-protein-coupled receptors (G-PCRs). Especially, EP1 is coupled to G_q_ protein alpha (Gαq), which activates phospholipase C (PLC), which, in turn, cleaves phosphatidylinositol 4,5-bisphosphate (PIP2), a membrane phospholipid, to generate second messenger’s inositol trisphosphate (IP_3_) and diacylglycerol (DAG). IP_3_ diffuses through the cytoplasm to bind to an open Ca^2+^ channel of the endoplasmic reticulum (sarcoplasmic reticulum in muscle tissue), increasing cytoplasmic Ca^2+^. DAG activates protein kinase C (PKC), which subsequently activates the mitogen-activated protein kinase (MAPK) pathway via extracellular signal–regulated kinase (ERK) phosphorylation [17,18].

Many studies have extensively investigated COX2 and PGES1 induction in response to acute inflammation [10,13,14,15,19,20]; however, few studies have reported which receptors in dermal fibroblasts recognize PGE2 and affect fibroblasts. This study investigated the role of EP1 in PGE2 signaling in normal human dermal fibroblasts (NHDFs).

## 2. Results and Discussion

### 2.1. Viability of PGE2 in NHDFs

Before investigating the messenger RNA (mRNA) expression of dermal aging-associated proteins in NHDFs after PGE2 treatment, I first determined the noncytotoxic PGE2 concentration using a Cell Counting Kit-8 (CCK-8) assay. PGE2-induced NHDF apoptosis gradually increased with the PGE2 concentration. Cell viability after 100 μg/mL PGE2 treatment decreased by 25.3% compared to the control (Figure 1A,B). In addition, phase contrast microscopy showed no significant differences in cell morphology after 50 μg/mL PGE2 treatment (Figure 1C). Therefore, I selected 50 μg/mL of PGE2 as the experimental condition for NHDFs in further experiments.

### 2.2. Effects of PGE2 on collagen, type I, alpha 1 (COL1A1) and MMP1 mRNA Expressions in NHDFs

Next, I assessed the effects of PGE2 on collagen, type I, alpha 1 (COL1A1) and MMP1 expression of NHDFs. PGE2 treatment reduced COL1A1 mRNA levels by ~61% and increased MMP1 mRNA levels by 2.1-fold (*p* < 0.05) (Figure 2A,B). Li et al. have shown that PGE2 is largely responsible for inhibiting procollagen expression in human skin organ culture [10]. The authors reported that increased COX2 expression in the aging dermis increases PGE2 production and consequently decreases collagen production. These results were consistent with my results, and I further confirmed that PGE2 treatment increases MMP1 expression. Nonsteroidal anti-inflammatory drugs (NSAIDs) are a class of commonly used drugs, such as aspirin, ibuprofen, diclofenac, and indomethacin, which reduce COX1 and COX2 activity, thereby decreasing PGE2 production [21]. Topical NSAIDs can decrease PGE2 production in human skin. These results provide a rationale for investigating the beneficial effects of NSAIDs for collagen deficiency or MMP1 excess in aging skin.

### 2.3. Effects of PGE2 on mRNA Expression–EP Receptors in NHDFs

With regard to PGE2-associated signaling transduction, PGE2 activity is exerted via four receptors: EP1–EP4 [22,23,24]. Skeletal muscles have muscle-specific stem cells (MuSCs) capable of tissue regeneration throughout life. PGE2, an inflammatory cytokine, targets MuSCs via EP4, leading to MuSC proliferation and regeneration [20]. With regard to fibrosis and keloids, recent studies have shown that EP2 transduces PGE2 signaling and results in collagen synthesis downregulation [25,26,27,28]. However, there are no studies on what PGE2 receptors are present in NHDFs. Therefore, to investigate PGE2 targets in NHDFs, the mRNA expression of EP1–EP4 were evaluated. First, I confirmed the downregulation of EP1 and EP4 gene/protein expressions by PGE2 treatment (Figure 3A,D,E). The mRNA expression of EP2 and EP3 was also evaluated. In contrast to EP1/4, I found no significant changes in EP2/3 expression after PGE2 treatment (Figure 3B,C). PGE2 treatment was expected to increase the expression of specific receptors between EP1–EP4 in NHDFs; however, I observed decreased EP1/4 expression, as shown in Figure 3A,D,E. Studies have shown that chronic insulin treatment decreases insulin receptor expression in adipose-derived insulin-producing cells [29], and a high insulin dose can induce alterations in insulin action, which might account, in part, for insulin-induced desensitization. PGE2 is a major eicosanoid product of fibroblasts and regulates fibroblast function in an autocrine fashion [30,31]. I believe that a similar result is observed in the decrease in EP1 expression after high-dose PGE2 treatment for 24 h.

### 2.4. EP1 siRNA Treatment Hides PGE2 Effects

To determine the role of EP1 in regulating the PGE2 reaction in NHDFs, I performed small interfering RNA (siRNA)-mediated EP1 knockdown experiments in NHDFs. Experiments were also performed by inhibiting the function of EP1–EP4 using antagonists or neutralizing antibodies. In the case of the receptor antagonist, I observed that the inhibition efficiency of each antagonist against PGE2 binding by EP1–EP4 was not constant. In the case of neutralizing antibodies, I found no commercialized antibodies that responded to EP1–EP4, so I applied the siRNA system to this study. When EP1 siRNA inhibited EP1 expression (Appendix A), I observed no significant change in mRNA/protein expressions of COL1A1/MMP1 between siRNA-transfected NHDFs and siRNA-transfected NHDFs with PGE2 (Figure 4). In contrast, after PGE2 treatment in EP2–EP4 knockdown conditions, mRNA expression of COL1A1 decreased and MMP1 expression increased (Appendix A). These results showed that EP1 is a receptor for PGE2 in NHDFs.

### 2.5. PGE2-Mediated Intracellular Signaling of EP1

EP receptors transmit signals by mobilization of intracellular Ca^2+^ and MAPK (EP1), increased cyclic adenosine monophosphate (cAMP; EP2 and EP4), or decreased cAMP (EP3). Because of these distinct signaling pathways, pleiotropic (or even antagonistic) responses to PGE2 might be seen in different cells, depending on the profile of the EP receptors expressed. Especially, EP1 signals through the G-PCR-mediated phosphatidylinositol-specific phospholipase C-β (PI-PLCβ) signaling pathway. PLCβ activation hydrolyzes PI 4,5-biphosphate into two second messengers, IP_3_ (which activates cytoplasmic Ca^2+^ levels via movement from endoplasmic reticulum stores, leading to activation of Ca^2+^ signaling pathways), and DAG (which triggers PKC). Subsequently, PKC triggers the MAPK pathway via Raf-1/ERK activation [17,18]. To confirm whether PGE2 binds to EP1 and affects intracellular signaling, I conducted time-course ERK1/2 phosphorylation. Results showed that ERK1/2 phosphorylation after 1, 2, and 4 h of PGE2 treatment significantly increased by ~2.5 times compared to the control (Figure 5A,B). However, I found no significant difference in p-ERK1/2 expression between control and stimulated conditions after ≥ 4 h of PGE2 treatment.

I also examined the effect of PGE2 treatment on intracellular Ca^2+^ levels using Fura-2-acetoxymethyl (Fura-2AM), a membrane-penetrating radiometric fluorescent indicator used for Ca^2+^ imaging by fluorescence. Inside the cells, the acetoxymethyl groups in Fura-2AM are removed by intracellular esterases, generating Fura-2, the pentacarboxylate Ca^2+^ indicator. The ratio of emissions at wavelengths of 340 and 380 nm is directly correlated to the cytoplasmic Ca^2+^ concentration [32,33,34]. Exposure of NHDFs to 50 μg/ml of PGE2 triggered a time-dependent increase in the intracellular Ca^2+^ concentration after 3 and 6 h of treatment (Figure 5C). Figure 5 shows how PGE2 induces intracellular signaling of EP1. Studies have reported EP1 agonists, such as ONO-8103, ONO-8359, GSK269984A, and GSK345931A, as EP1 inhibitors [35,36,37,38]. Each inhibitor alleviates bladder pain [35] or suppresses acid-derived heartburn symptoms [36] and inflammatory pain [38]. Future studies are required in order to identify these agonists as potential anti-aging materials.

In summary, PGE2 decreases collagen type I expression and increases MMP1 expression [10]; inhibition of collagen expression and MMP1 promotion by PGE2 are skin aging mechanisms. PGE2 also decreases EP1 expression in NHDFs. However, when treated under conditions of EP1 knockdown in NHDFs, PGE2 does not affect collagen type I and MMP1 expression. In addition, PGE2 bound to EP1 transmits the signal through intracellular Ca^2+^ and ERK pathway. NHDFs are used as a model to evaluate transcriptional changes in PGE2 treatment, demonstrating that EP1 plays an important role in PGE2 signaling. These results suggest that PGE2 is directly associated with the EP1 receptor pathway-regulated ERK1/2 and IP_3_ signaling in NHDFs. This study provided a rationale for the beneficial effects of blocking EP1–PGE2 interaction on collagen deficiency in aging skin (Figure 6).

## 3. Materials and Methods

### 3.1. Cell Culture of NHDFs

NHDFs (Lonza, Basel, Switzerland) were cultured in Dulbecco’s Modified Eagle Medium (DMEM; Welgene, Gyeongsan, Korea) containing 10% fetal bovine serum (FBS; Equitech-Bio, Kerville, Texas, USA), 100 U/mL of penicillin, and 100 μg/mL of streptomycin in a 5% CO_2_ humidified incubator at 37 °C. The NHDFs were used for experiments at passages three–12.

### 3.2. Effects of PGE2 on COL1A1 and MMP1 mRNA Expressions in NHDFs

To assess the effects of PGE2 on COL1A1 and MMP1 expression of NHDFs, NHDFs were cultured in serum-free DMEM for 24 h in the presence of PGE2 or dimethyl sulfoxide (DMSO) (Sigma-Aldrich Corporation, St. Louis, MO, USA) as the control. COL1A1 and MMP1 expression was examined by quantitative reverse transcription polymerase chain reaction (qRT-PCR).

### 3.3. In Vitro Cell Viability Assay

The cell viability of NHDFs was determined using a CCK-8 assay (Dojindo, Kumamoto, Japan). Briefly, NHDFs were seeded in 96-well tissue culture plates at a density of 2 × 10^3^ cells per well, treated with 10 μL of CCK-8 solution in 90 μL of phenol red–free DMEM (Welgene), and incubated for 1 h at 37°C. Absorbance was measured at a wavelength of 450 nm using an Epoch microplate spectrophotometer (BioTek Instruments, Inc., Winooski, VT, USA).

### 3.4. Real-Time qRT-PCR

Total RNA was extracted from NHDFs using TRIzol^®^ reagent (Invitrogen, Carlsbad, CA, USA), and the RNA concentration was quantified using an Epoch Take3 microvolume spectrophotometer (BioTek Instruments). Then, 2 μg of RNA was reverse-transcribed into complementary DNA (cDNA) using a Superior Script III master mix (Enzynomix, Daejeon, Korea), and reverse transcription was stopped by adding tris–ethylenediaminetetraacetic acid (EDTA) buffer (pH 8.0) to 100 μL of cDNA solution. Finally, qRT-PCR was performed using a StepOnePlus^TM^ Real-Time PCR System (Thermo Fisher Scientific, Waltham, MA, USA) according to the manufacturer’s instructions. Briefly, 20 μL of real-time PCR mixture contained 10 μL of 2X TaqMan Universal PCR Master Mix II, 50 ng of cDNA, and 1 μL of 20X TaqMan Gene Expression assay^®^ solution (Applied Biosystems, Foster City, CA, USA). Table 1 presents the gene identification numbers for the TaqMan Gene Expression assay used in real-time qRT-PCR analysis. Human glyceraldehyde 3-phosphate dehydrogenase (GAPDH) was used to normalize variation in cDNA quantities from different samples.

### 3.5. EP1 siRNA Transfection

After they reached ~70%–80% confluency, NHDFs were transfected with an EP1-specific silencer^®^ select siRNA (S194725; Thermo Fisher Scientific) or a negative control siRNA (Thermo Fisher Scientific). Transfection was performed using Lipofectamine^®^ RNAiMAX^TM^ (Thermo Fisher Scientific) according to the manufacturer’s instructions. Briefly, a mixture of human EP1 siRNA and Lipofectamine^®^ RNAiMAX^TM^ was diluted in 500 μL of antibiotic-free Opti-MEM^TM^ (Invitrogen). After slightly agitating the mixture for 20 min at room temperature, NHDFs were incubated in the mixture for 6 h at 37 °C. Then, the medium was removed and exchanged with DMEM containing 10% FBS. Table 2 presents siRNA identification numbers for the gene knockdown assay.

### 3.6. Immunoblotting Analysis

NHDFs were lysed on ice for 2 h with radioimmunoprecipitation assay (RIPA) lysis buffer (Merck Millipore, Burlington, MA, USA) with a protease/phosphatase inhibitor cocktail (Merck Millipore). After incubation, NHDF lysates were centrifuged at 12,000 rpm for 15 min at 4 °C, and the supernatants were collected into new EP-tubes. For immunoblotting analysis, 20 μg of protein was separated on 4%–12% gradient Bis-Tris gels before transfer to polyvinylidene difluoride (PVDF) membranes (Thermo Fisher Scientific). The PVDF membranes then were blocked by incubation in Tris-buffered saline with Tween-20 (TBST) buffer containing 1% bovine serum albumin (BSA; Sigma-Aldrich) for 1 h at room temperature and incubated overnight at 4 °C with the GAPDH primary antibody (1:1,000; Santa Cruz Biotechnology, Dallas, TX, USA), Collagen I primary antibody (1:1,000, Abcam, UK), MMP1 primary antibody (1:1,000, Abcam, UK), EP1 primary receptor (1:1,000, Abcam, Cambridge, UK), EP4 primary receptor (1:300, Abcam, UK), ERK1/2 primary antibody, or the phosphor-ERK1/2 primary antibody (1:1,000; Abcam, UK) in TBST containing 1% BSA. Blots were washed thrice with TBST and incubated with horseradish peroxidase (HRP)-conjugated secondary antibodies (1:3,000; Bio-Rad Laboratories, Inc., Hercules, CA, USA) for 1.5 h at room temperature. Finally, the blots were developed using Clarity Western ECL blotting substrate (Bio-Rad Laboratories) according to the manufacturer’s instructions.

### 3.7. Ca^2+^ Imaging

Fura-2-AM (Abcam) is a membrane-penetrating derivative of the radiometric Ca^2+^ indicator used to measure cytoplasmic Ca^2+^ concentrations by fluorescence. Ca^2+^ imaging of Fura-2AM-loaded NHDFs was performed using a CELENA^®^ S digital imaging fluorescence microscope (Logos Biosystems, Daejeon, Korea) with excitation at a wavelength of 405 nm and fluorescence emission measurements at a wavelength of 519 nm. NHDFs on the confocal dish (SPL, Pocheon, Korea) were loaded for 30 min in the dark at room temperature in Hanks’ Balanced Salt Solution (HBSS) (Welgene) containing 2 μM Fura-2AM. The dish was then rinsed thrice with HBSS. Next, the NHDFs were again incubated for 30 min at 37 °C. Finally, they were washed with prewarmed HBSS, and the live NHDFs were imaged using a fluorescence microscope.

### 3.8. Statistical Analysis

Statistical analysis of data was performed using one-way analysis of variance (ANOVA). Results were presented as the mean ± standard deviation (SD) of three or more independent experiments. *p* < 0.05 was considered statistically significant.

## Figures and Tables

**Figure 1 ijms-20-05555-f001:**
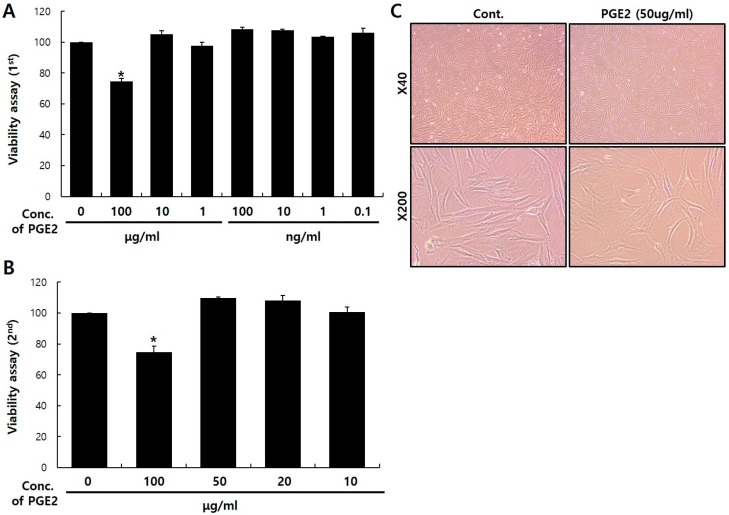
Viability and characterization of prostaglandin E synthases 2 (PGE2) treatment in normal human dermal fibroblasts (NHDFs). NHDFs were seeded in 96-well plates at a density of 2 × 10^3^ cells per well and treated with PGE2 (**A**,**B**). Cell viability results, based on a Cell Counting Kit-8 (CCK-8) assay, are presented as the mean ± standard deviation (SD) of the percentage of the control optical density (OD). (**C**) NHDF morphologies were visualized with phase contrast microscopy (×40, ×200). NHDFs were treated with 50 μg/mL of PGE2 or dimethyl sulfoxide (DMSO) for 24 h. Values represent the mean ± SD of three independent experiments. * *p* < 0.05 compared to the control.

**Figure 2 ijms-20-05555-f002:**
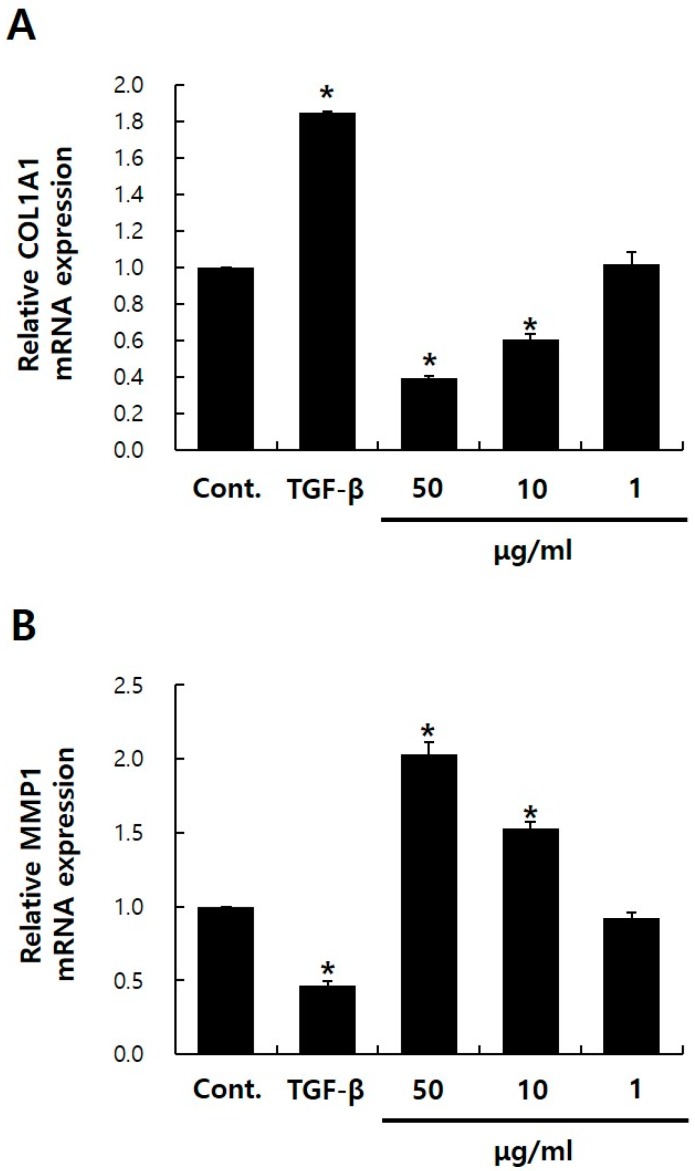
Characterization of PGE2 treatment in NHDFs. Real-time quantitative reverse transcription polymerase chain reaction (qRT-PCR) analysis of dermal fibroblast markers collagen, type I, alpha 1 (COL1A1) (**A**) and matrix metallopeptidase 1 (MMP1) (**B**). transforming growth factor beta (TGF-β) was used as a positive control. Values represent the mean ± SD of three independent experiments. * *p* < 0.05 compared to the control. Control means NHDFs treated with DMSO.

**Figure 3 ijms-20-05555-f003:**
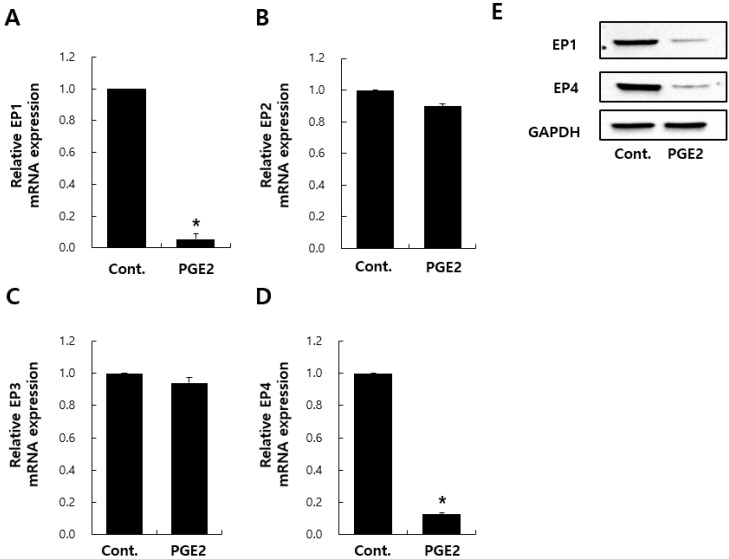
Analysis of transcript levels of different receptors E-prostanoid 1–4 (EP1–EP4) after PGE2 treatment. Transcriptional expression of PGE2 receptors (EP1–EP4) by NHDFs after 24 h treatment with DMSO or PGE2 (**A**–**D**). Immunoblotting analysis for EP1 and EP4 (**E**). Values represent the mean ± SD of three independent experiments. * *p* < 0.05 compared to the control.

**Figure 4 ijms-20-05555-f004:**
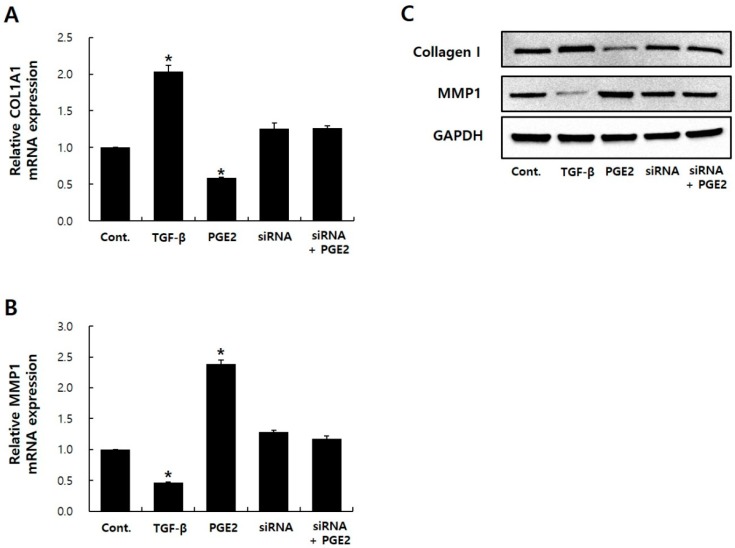
EP1 siRNA blocks PGE2 effects on COL1A1 and MMP1 expression in NHDFs. COL1A1 (**A**) and MMP1 (**B**) expression was estimated by qRT-PCR. Immunoblotting analysis for Collagen I and MMP1 (**C**). TGF-β was used as a positive control. The error bar represents the SD of independent experiments. * *p* < 0.05 compared to the control. siRNA, small interfering RNA.

**Figure 5 ijms-20-05555-f005:**
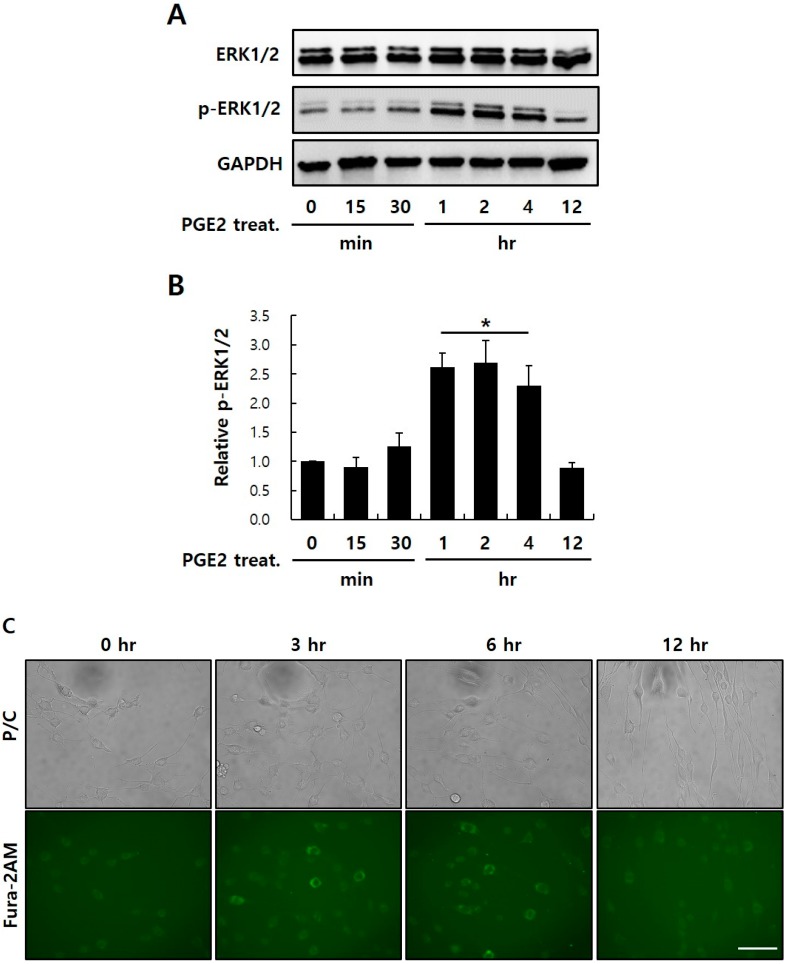
Determining the signaling pathway regulating PGE2–EP1 interaction in NHDFs. (**A**) Immunoblotting analysis for proteins associated with the mitogen-activated protein kinase (MAPK) signaling pathway. NHDFs were treated with 50 μg/mL of PGE2 for various time courses. (**B**) Relative intensity values for p-extracellular signal–regulated kinase (ERK)1/2 were densiometrically quantified using Multi-Gauge V3.0 (Fujifilm Inc., Japan). Glyceraldehyde 3-phosphate dehydrogenase (GAPDH) was used to normalize variation in protein quantities from different samples. (**C**) Effects of PGE2 on the increase in intracellular Ca^2+^ concentration in a time-dependent manner in NHDFs, as measured by immunocytochemistry, where 2 uM Fura-2AM was used as an indicator of cytoplasmic Ca^2+^ (× 200). Results are the mean ± SD of three independent experiments. * *p* < 0.05 compared to the control. Scale bar: 30 μm

**Figure 6 ijms-20-05555-f006:**
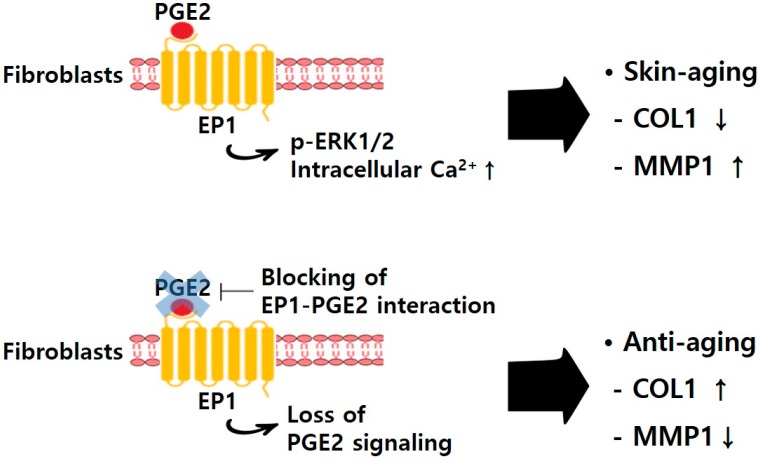
Schematic model for the mechanism contributing to beneficial effects of blocking EP1–PGE2 interaction in NHDFs. ↑, increase; ↓, decrease.

**Table 1 ijms-20-05555-t001:** Gene Name and Assay ID Number in Real-time RT-PCR Analysis.

Symbol	Gene Name	Assay ID
COL1A1	Collagen, type I, alpha 1	Hs00164004_m1
MMP1	Matrix metallopeptidase 1 (interstitial collagenase)	Hs00899658_m1
EP1	Prostaglandin E receptor 1	Hs00168752_m1
EP2	Prostaglandin E receptor 2	Hs00168754_m1
EP3	Prostaglandin E receptor 3	Hs00168755_m1
EP4	Prostaglandin E receptor 4	Hs00168761_m1
GAPDH	Glyceraldehyde-3-phosphate dehydrogenase	43333764F

**Table 2 ijms-20-05555-t002:** siRNA Name and siRNA ID number in knockdown analysis.

Symbol	ID Number
EP1 siRNA	S194725
EP2 siRNA	S11448
EP3 siRNA	S11451
EP4 siRNA	S11454
Control siRNA	D-001810-01-05

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
