# Peer review of "Prostaglandin E2 Induces Skin Aging via E-Prostanoid 1 in Normal Human Dermal Fibroblasts"

_ijms, 2019, doi:10.3390/ijms20225555_

Round 1

Reviewer 1 Report

Manuscript by JH Shim described the relationship of PGE2-EP1 in the expression of markers of skin aging (COL1A1, MMP1).

There are some aspects that must be improved:

- Fig 1A. It has not been specified which product corresponds to each panel of concentrations   

- Fig. 1B. Show changes of morphology of NHDFs at higher concentrattion of PGE2

- 2.2. “Effects of PGE2 on mRNa expression–associated skin sginbg in NHDFs” Author study the expression of COL1A1 and MMP1, thus the title of section 2.2 must say “Effects of PGE2 on COL1A1 and MMP1 mRNa expression in NHDFs” Correct also the corresponding Material and Methods section.  

- Fig. 2. Specifyin the legend of the figure if the column “Control” corresponds to untreated cells, cells treated with DMSO...

Author Response

Manuscript by JH Shim described the relationship of PGE2-EP1 in the expression of markers of skin aging (COL1A1, MMP1).

There are some aspects that must be improved:

- Fig 1A. It has not been specified which product corresponds to each panel of concentrations   

Answer: I appreciate the reviewer’s close investigation on my manuscript. I marked it in the figure 1A.

- Fig. 1B. Show changes of morphology of NHDFs at higher concentrattion of PGE2.

Answer: I thank for the thoughtful suggestion from the reviewer. At over 100 ug/ml of PGE2 concentration, the cells died and became floating in the culture medium, so I could not present cell morphology.

- 2.2. “Effects of PGE2 on mRNA expression–associated skin aging in NHDFs” Author study the expression of COL1A1 and MMP1, thus the title of section 2.2 must say “Effects of PGE2 on COL1A1 and MMP1 mRNa expression in NHDFs” Correct also the corresponding Material and Methods section.  

Answer: I appreciate the reviewer’s close investigation on my manuscript. I corrected it according to reviewer`s instruction.

- Fig. 2. Specify in the legend of the figure if the column “Control” corresponds to untreated cells, cells treated with DMSO...

Answer: I thank for the thoughtful suggestion from the reviewer. I have modified the figure legend according to reviwer`s instruction.

Reviewer 2 Report

This manuscript describes that prostaglandin E2 induces skin aging through EP1 receptor in normal human dermal fibroblasts. This story might have some interest in aging biology, but several points need to be addressed to validate their findings.

For figure-2, (A) Author has mentioned in the results section that COL1A1 mRNA is decreased by ~79% but figure shows ~60% decrease in 50 ug/ml PGE-2 concentration (value at Cont is ~1.0 and value at 50 ug/ml is ~0.4) and also author placed significance bar between 50 ug/ml and 10 ug/ml concentration but not sure why? It should be between Cont and PGE-2 treatment. In panel (B), I see an almost similar issue like panel (A). 50 ug/ml PGE-2 concentration shows ~2-fold increase (cont ~1.0 vs 50 ug/ml ~2.0) but 4.4-fold mentioned in the results section. Similarly, the significance bar has the same issue as (A). The author has primarily concluded the story based on mRNA expression. The author should validate all the mRNA expression by western blot ( COL1A1 and MMP1 for figure 2 and figure 4) The author mentioned that EP1 and EP4 are decreased upon PGE-2 treatment and provided insulin receptor as an example, so, the author must add western blot to validate this data. Taqman primers were used for the qPCR but not sure what type of GAPDH primer (43333764F) author used here; could find this catalog number for GAPDH Taqman primer. Figure 1 could be used as a supplemental figure which is used to show the nontoxic dose of PGE-2 for fibroblast. The author should add label just below the western blot in figure 5(A), it’s hard to follow the label from 5(B) and they should also add PGE-2 with this label.

Author Response

Reviewer 2

This manuscript describes that prostaglandin E2 induces skin aging through EP1 receptor in normal human dermal fibroblasts. This story might have some interest in aging biology, but several points need to be addressed to validate their findings.

For figure-2, (A) Author has mentioned in the results section that COL1A1 mRNA is decreased by ~79% but figure shows ~60% decrease in 50 ug/ml PGE-2 concentration (value at Cont is ~1.0 and value at 50 ug/ml is ~0.4) and also author placed significance bar between 50 ug/ml and 10 ug/ml concentration but not sure why? It should be between Cont and PGE-2 treatment.

Answer: I appreciate the reviewer’s close investigation on our manuscript. First, I corrected “79%” into “61%”. Second, * means between control and each conditions. I ug/ml of PGE2 does not effect on COL1A1 and MMP1 expressions. And I corrected figure 2 clearly not to be confused.

In panel (B), I see an almost similar issue like panel (A). 50 ug/ml PGE-2 concentration shows ~2-fold increase (cont ~1.0 vs 50 ug/ml ~2.0) but 4.4-fold mentioned in the results section. Similarly, the significance bar has the same issue as (A).

Answer: I appreciate the reviewer’s close investigation on our manuscript. I corrected “4.4-fold” into “2.1-fold”.

The author has primarily concluded the story based on mRNA expression. The author should validate all the mRNA expression by western blot ( COL1A1 and MMP1 for figure 2 and figure 4)

Answer: I thank for the thoughtful suggestion from the reviewer. The effects of COL1A1 and MMP1 by PGE2 (10th and 21st references mentioned in my manuscript) were reported already. So I checked the mRNA expressions and moved to next figures. Please understand this circumstance.

The author mentioned that EP1 and EP4 are decreased upon PGE-2 treatment and provided insulin receptor as an example, so, the author must add western blot to validate this data.

Answer: I appreciate the reviewer’s close investigation on our manuscript. I can`t comment on this paper, however I`m working on a follow-up study. Please understand that I will be able to show reviewer the results of the reviewer`s comments in subsequent my paper.

Taqman primers were used for the qPCR but not sure what type of GAPDH primer (43333764F) author used here; could find this catalog number for GAPDH Taqman primer.

Answer: I thank for the thoughtful suggestion from the reviewer.

https://www.thermofisher.com/order/catalog/product/4352934E?SID=srch-hj-4352934E

To help reviewer`s understading, I included the address and screenshot of the GPADH product site. I used “4333764F” GPADH.

Figure 1 could be used as a supplemental figure which is used to show the nontoxic dose of PGE-2 for fibroblast.

Answer: I appreciate the reviewer’s close investigation on our manuscript. Another reviewer inqured about the figure 1, so I would like to place it as Fig. 1. Please understand.

The author should add label just below the western blot in figure 5(A), it’s hard to follow the label from 5(B) and they should also add PGE-2 with this label.

Answer: I thank for the thoughtful suggestion from the reviewer. I corrected Fig 5. as reviewer said.

Round 2

Reviewer 2 Report

In the initial comments, I asked the author to validate Fig-2 and Fig-4 data by western blot; the author has cited two references (Ref-10 and Ref-21) about the protein expression (COL1A1 and MMP1) after PGE2 treatment. I was going through those references and found that Ref-10 (Fig-6a) used only 10nM PGE2 concentration for COL1A1 western blot whereas this paper is using 50 ug/ml (~142000nM; 14200 times more than the ref-10 if I'm not mistaken); how can these two data (10nM vs 142000nM) be compared? I request the author to mention specific locations (results/figures in the specific references) of MMP1 western blot after PGE2 treatment. Again, it seems that these data are the backbone of this manuscript, so protein expression is very important for this paper. Also, protein expression for COL1A1 and MMP1 after siRNA (at least EP1) treatment (with or without PGE2) would be an important piece of data for this manuscript.

The author wanted to do the follow-up work for the EB1 and EB4 WB upon PGE2 treatment which does not justify the limitations of this paper. Estimation of transcript level (mRNA) and protein level both are very important for the mechanism-based study.

Author Response

In the initial comments, I asked the author to validate Fig-2 and Fig-4 data by western blot; the author has cited two references (Ref-10 and Ref-21) about the protein expression (COL1A1 and MMP1) after PGE2 treatment. I was going through those references and found that Ref-10 (Fig-6a) used only 10nM PGE2 concentration for COL1A1 western blot whereas this paper is using 50 ug/ml (~142000nM; 14200 times more than the ref-10 if I'm not mistaken); how can these two data (10nM vs 142000nM) be compared? I request the author to mention specific locations (results/figures in the specific references) of MMP1 western blot after PGE2 treatment. Again, it seems that these data are the backbone of this manuscript, so protein expression is very important for this paper. Also, protein expression for COL1A1 and MMP1 after siRNA (at least EP1) treatment (with or without PGE2) would be an important piece of data for this manuscript. The author wanted to do the follow-up work for the EP1 and EP4 WB upon PGE2 treatment which does not justify the limitations of this paper. Estimation of transcript level (mRNA) and protein level both are very important for the mechanism-based study.

Answer: I appreciate the reviewer’s close investigation on my manuscript. Wester data for Fig. 4 contains Fig. 2. So I conducted western blot and added to Fig. 4.

(Western images are included in attached file.)

Round 3

Reviewer 2 Report

The author addressed all comments.